# Anticancer Therapeutic Effects of Green Tea Catechins (GTCs) When Integrated with Antioxidant Natural Components

**DOI:** 10.3390/molecules28052151

**Published:** 2023-02-24

**Authors:** Jae-Wook Oh, Manikandan Muthu, Suraj Shiv Charan Pushparaj, Judy Gopal

**Affiliations:** 1Department of Stem Cell and Regenerative Biology, Konkuk University, Seoul 05029, Republic of Korea; 2Department of Research and Innovation, Saveetha School of Engineering, Saveetha Institute of Medical and Technical Sciences (SIMATS), Thandalam, Chennai 602105, India

**Keywords:** green tea catechins, antioxidant, natural compounds, anticancer, mechanisms, apoptosis

## Abstract

After decades of research and development concerning cancer treatment, cancer is still at large and very much a threat to the global human population. Cancer remedies have been sought from all possible directions, including chemicals, irradiation, nanomaterials, natural compounds, and the like. In this current review, we surveyed the milestones achieved by green tea catechins and what has been accomplished in cancer therapy. Specifically, we have assessed the synergistic anticarcinogenic effects when green tea catechins (GTCs) are combined with other antioxidant-rich natural compounds. Living in an age of inadequacies, combinatorial approaches are gaining momentum, and GTCs have progressed much, yet there are insufficiencies that can be improvised when combined with natural antioxidant compounds. This review highlights that there are not many reports in this specific area and encourages and recommends research attention in this direction. The antioxidant/prooxidant mechanisms of GTCs have also been highlighted. The current scenario and the future of such combinatorial approaches have been addressed, and the lacunae in this aspect have been discussed.

## 1. Introduction

Cancer is a multifactorial disorder that typically arises through the influence of diverse genetic and environmental factors [1]. Cancer claimed nearly 10 million lives in 2020, the highest number of deaths through a specific disease [2]. One out of six mortalities has been statistically proven to be due to cancer. Furthermore, it is predicted that the number of global morbidities due to cancer by 2025 will number around 20 million [3,4]. Even with the advent of cutting-edge technologies for generating cancer models, genetic networks, and molecular and cellular interactions in cancer models revealing novel targets for cancer therapies [5], the disease is still at large. Despite the fact that enormous progress has been made in identifying drug targets and therapeutic molecules for cancer cell treatment, the drug resistance of cancer cells against the currently available anticancer drugs is a major bottleneck in cancer therapy [6]. In addition to that, cancer stem cells [7], insufficient bioavailability [8], and side effects of anticancer drugs [9] are also major setbacks in the progress of cancer treatments. These factors instigate the urge to explore alternative novel treatment strategies to overcome the hurdles of the present cancer therapies.

Molecules from a wide range of natural bioresources such as bacteria, fungi, plants, protozoa, and animals possess cancer therapeutic potential [10]. Among the natural products, Camptothecin and Taxol, which were initially isolated from the bark of *Camptotheca acuminata* and also from many fungal species, have been well-proven to be effective against many cancers and are under human clinical trials [11,12,13,14,15]. In addition to that, various other plant-derived compounds also exhibit anticancer activity, for instance, hematoxylin from the heartwood of *Haematoxylon campechianum* [16], Eucalyptin A from the fruits of *Eucalyptus globulus* [17], Pseudolaric acid B from the root bark of *Pseudolarix kaempferi* [18,19], Parthenolide, a sesquiterpene lactone, from the shoots of *Tanacetum parthenium* [20], exhibited anticancer activity by acting on different cancer targets [21].

Similarly, numerous natural compounds that exhibit anticancer potential have also been discovered in marine resources. Among these, the very first marine organism that gave a glimpse of an anticancer compound (arabino nucleosides and cytosine arabinose) was the Caribbean sponge, *Cryptotethya crypta* [20]. Despite natural compounds being constantly explored for anticancer potential from all sources and some being already approved as drug candidates for chemotherapy, the WHO specifically suggested that chemoprevention is the most effective therapy for various cancers, which include breast malignancies [22].

Therefore, screening of natural compounds with chemopreventive properties has been widely carried out. For example, ethanolic extracts of *Arenosclera brasiliensis* [23] and *Haliclona koremella* [24], marine sponges, were identified with chemoprevention properties. Further, the antioxidants, namely verongiaquinol and manzamine A, of the Red Sea sponge *Aplysina* sp. and *Acanthostrongylophora* sp., respectively, also exhibited antimutagenic potential [25]. Carotenoids, a group of natural compounds found across most forms of life, such as bacteria, algae, fungi, plants, and animals, have proven to have tremendous antioxidant potential and are also used as chemopreventive agents [26].

Tea is yet another important source of innumerable natural compounds with chemopreventive capacity. Among the various variants of teas, green tea is the best-studied system for its cancer chemopreventive and chemotherapeutic effects [27,28]. It has been both approved and proven that green tea reduces the risk of breast, prostate, thyroid, colorectal, stomach, esophageal, and prostate cancers [29,30,31,32,33,34]. The chemopreventive activity of green tea is mediated through catechins (flavan-3-ols), and the major bioactive component of the green tea extracts are: (−)-epicatechin (EC), (−)-epigallocatechin (EGC), (−)-epicatechin-3-gallate (ECG), and (−)-epigallocatechin-3-gallate (EGCG) [35] among which EGCG is the most abundant and biologically active [36].

Among the other catechins, EGCG is well known for its inhibitory activity at all stages of cancer initiation, promotion, and progression. Other catechins, such as ECG and EGC, are relatively lower. EGCG inhibits protein kinases affecting cell growth and activating protein kinases linked to cell apoptosis and suppresses proteinases such as matrix metalloproteinase (MMPs) via inhibition of cancer cell migration, invasion, and metastasis. In addition, EGCG possesses antioxidant, anti-inflammatory, antiproliferative, antiangiogenic, and anti-metastatic effects by modulating signaling pathways, enzymatic activity, and protein kinases [37,38]. Figure 1 enlists the various anticarcinogenic mechanisms of GTCs. However, in humans, the plasma bioavailability of GTCs is very low, which has been in part attributed to their oxidation, metabolism, and efflux [39,40,41,42].

In the present review, we have broadly reviewed the biological benefits of green tea catechins, and the specific anticancer applications and the milestones achieved through green tea catechins have been consolidated and presented. Finally, the main objective of this review is to highlight the combinatorial effect of catechins combined with various natural compounds. The future of such a combinatorial approach has been presented, and the lapses in the current knowledge have been addressed.

## 2. Biological Activity of Green Tea Catechins

Along with amino acids, proteins, lipids, and catechins, also known as flavan-3-ols, are a class of naturally occurring phenol/polyphenolic chemicals that are found in *Camellia sinesis* [43,44]. Until now, 8 different catechins, including C ((−)-catechin, EC ((−)-epicatechin), ECG ((−)-epicatechingallate), EGC ((−)-epigallocatechin), EGCG ((−)-epigallocatechin gallate), GC ((−)-gallocatechin), CG ((−)-catechingallate), and GCG ((−)-gallocatechin gallate), have been extracted from green tea, of which EC, EGC, ECG and EGCG are the dominant ones [45,46,47]. The biological properties of catechins include: antioxidants, anti-tumor, anti-inflammatory, anti-microbial, anti-viral, anti-diabetic, anti-obesity, and hypotensive effects [48,49,50,51].

These properties of green tea are found helpful for the treatment of obesity, diabetes, cardiovascular disease (CVD), nervous problems, and oral hygiene. For instance, catechins have been shown to regulate cell growth and nourishment and cause programmed cell death of tumor cells [50,52,53,54]. Numerous ailments and diseases, such as aging, arthritis, cancer, cardiovascular disease, diabetes, and obesity, include inflammation as the common component. The ability to reduce protein denaturation and boost the generation of anti-inflammatory cytokines are two of green tea’s overall anti-inflammatory capabilities [55]. Oxidative stress resulting from ROS has serious health implications, as it can damage DNA, affect protein folding, and reduce the ability of the mitochondria to produce ATP. As a result, the brain’s cognitive abilities may decline and may lead to conditions such as Alzheimer’s and Parkinson’s. Green tea’s anti-inflammatory and antioxidant qualities help protect neurons, and studies have demonstrated that its metabolites can pass the blood-brain barrier [56,57,58,59,60]. By binding to ROS, green tea’s antioxidant nature help neutralize free radicals inside the body and upregulates the activity of antioxidant enzymes [61,62]. Likewise, green tea’s anti-inflammatory and antioxidant properties have an impact on treating CVD. Additionally, regular intake of green tea has become instrumental in preventing atherosclerosis, lower total cholesterol levels, and enhancing the low-density lipoprotein (LDL) to high-density lipoprotein (HDL) ratio [63,64]. Specific symptoms, such as increased waist circumference, higher plasma triglycerides, increased LDL/HDL ratio, raised fasting blood glucose, and elevated blood pressure, are closely linked to metabolic syndromes such as diabetes and obesity [65,66]. Insulin resistance and perhaps reduced insulin production are additional features of type 2 diabetes. Green tea has been demonstrated to boost glucose-induced insulin production and improve receptor sensitivity for insulin [67,68]. Green tea has been demonstrated to decrease blood pressure and control obesity by improving HDL and reducing LDL, triglycerides, and body waist circumference by inhibiting digestive enzymes and fat absorption [69,70,71].

According to published research, green tea is proven to be antibacterial against a majority of oral microorganisms. Additionally, it has been demonstrated to enhance oral health by raising oral peroxidase activity, delaying the onset and progression of periodontitis, lowering dentin erosion and tooth loss, and thus contributing to the reduction of bad breath [72,73,74,75,76]. By interacting with the Pneumolysin and Sortase A of *Streptococcus pneumoniae*, epigallocatechin gallate (EGCG) was discovered to regulate antibiotic resistance [77]. Green tea was discovered to have a far higher level of antibacterial activity packed in the nano-sized particles that diffuse out from teabags compared to micro- and macro-sized particles. The bactericidal action was due to the abundance and higher active surface area of the catechins [78]. In a 2015 study, Deepak Kumar et al. examined the antimicrobial properties of 12 synthetic derivatives of catechins. Three of these derivatives of catechins exhibited antibacterial activity, and one of these compounds also demonstrated high antifungal activity. The docking investigation evidenced the catechins’ affinity for the ATP-binding region of DNA gyrase as the driving force behind the antibacterial or antifungal effect. There has been extensive research on the antibacterial properties of GTCs on microbes. It has been demonstrated that green tea can fight these germs both directly and indirectly and that it can also work synergistically along with routine antibiotics. The anti-inflammatory and antioxidant properties of green tea, together with other well-known health advantages, may also aid in the antibacterial effects. Studies on *Escherichia coli* revealed that exposure to green tea polyphenols (GTPs) caused significant changes in the expression of 17 genes, with nine genes being upregulated and eight genes being downregulated [79,80,81].

The presence of many structural -OH groups invest strong antioxidant property of the GTCs. More than ten families of chemicals make up the chemical formulation of green tea. Phenolic acids, polyphenolic compounds, amino acids, and lipids make up the primary constituents of green tea [43,44,82,83,84,85]. Due to the beneficial aspects of GTCs mentioned above, it has been recommended as a dietary product on a daily basis [86].

## 3. Anticancer Activity of Green Tea Catechins

Most of green tea’s anti-tumor properties are catechin-regulated, with EGCG having the strongest impact. EGCG has the highest inhibitory activity, followed by ECG > EGC > EC. Moreover, combinations of catechins have shown enhanced anti-tumor activity than isolated EGCG due to their combined effect. According to Fujiki et al. [87], GTCs have a variety of anti-mutagenic and anticarcinogenic effects on human malignancies, including those of the breast, esophagus, colon, prostate, small intestine, stomach, lung, and liver. Several of these effects will be discussed in this section.

It is believed that green tea compounds, GTC mixtures, or pure EGCG are capable of influencing the carcinogenesis cycle, tumor origin, proliferation, and growth. This has been realized based on investigations using animal models and cancer cell lines [86]. For in-vivo investigations, xenograft tumor models of injecting human tumor cells subcutaneously into naked mice were conducted. Tumors typically develop over time depending on the cell concentrations injected. Mice were administered EGCG via intraperitoneal injection, dietary water or feed, and oral gavage, with varying concentrations of catechins consumed for the entire treatment in accordance with various experimental designs. Though the dosage of GTCs is varied with respect to the type of cancer cells studied, relatively fixed concentrations of EGCG (5–200 µM) were used predominantly. GTCs have been proven to reduce telomerase in cell lines, induce cell death, arrest cell cycles, and beneficially target cell receptors through binding with receptor tyrosine kinases (RTKs) [88]. GTCs have the capacity to both neutralize and generate ROS [89]. The antioxidant ability of GTCs has been ranked in the order ECG > EGCG > EGC > EC [90], and the prooxidant activity of GTCs has been reported to be crucial for programmed cell death and suppression of cancer cell development. GTCs are good demethylating agents and can be used as epigenetic modifiers to change the histone and the transcription of miRNAs in order to epigenetically control cellular processes and inhibit oncogene expression [91,92]. It has been established that GTCs, such as EGCG, can inhibit cancer growth in numerous mechanisms involving many molecular cues. Initially, EGCG binds with one or more target proteins, in this case usually a transmembrane receptor such as kinases, and modifies the regulation of signaling and metabolic processes that are crucial for the growth of cancerous cells. Figure 2 gives an overview of the various cancer-related processes that GTCs affect and interplay with.

Breast cancers are the most prevalent cancers among women around the world and appear due to many factors [93]. Epidemiological research in China has demonstrated that drinking green tea has positive effects on breast cancer prevention and recurrence, particularly for women who drink more than 4 cups of tea each day [94,95,96]. Numerous researchers have examined the mechanism through which catechins inhibit malignancy. Tea catechins, such as EGCG, ECG, and EC, have a potent antiangiogenic impact that inhibits cell proliferation and triggers apoptosis of breast cancer cells by neutralizing reactive oxygen species (ROS)-induced oxidative stress [97,98]. This was supported by reports validating the effect of EGCG on breast cancer cell inhibition and migration by down-regulating the phosphoinositide 3-kinases (PI3K)/Akt (Protein kinase B) and tumor protein p53/B-cell lymphoma (Bcl)-2 signaling pathways as well as modification of telomerase [99,100]. Additionally, prior research has demonstrated that tea catechins affect cell membrane receptors and restrict the spread of breast cancer cells by lowering levels of neo-nourishing factors such as vascular endothelial growth factor (VEGF) and epidermal growth factor (EGFR) and blocking the chemical signaling and activation of protein transcription-3 (STAT-3) and nuclear factor kappa B (NF-κB) [101,102]. Moreover, EGCG can inhibit breast cancer progression by hindering the focal adhesion kinase (FAK) signaling pathway, binding target proteins such as estrogen receptors (ERs) [101,103]; their antiproliferative activity by blocking ERβ-specific inhibitor is also reported [104]. Additionally, it has been suggested that EGCG modulates signal peptide-CUB-EGF domain-containing protein 2 (SCUBE2) gene expression in breast cancer cells and DNA methyltransferases [105]. Deb and colleagues typically found that 20 μM EGCG reduced the expression of the epigenetically suppressed TIMP-3 gene [106]. Additionally, ER + progesterone receptors (PR) + cancer cells were used to evaluate the anticancer effects of EGCG by epigenetic downregulation of ER- through p38 mitogen-activated protein kinase(p38MAPK)/casein kinase 2 (CK2) activation [107]. Animal models have demonstrated that administering EGCG inhibits the growth of cancerous breast tissue. The observations in a xenograft model revealed that EGCG inhibited pro-tumor macrophage invasion and macrophage 2 (M2) polarization after intraperitoneal administration of 10 mg/kg EGCG [108]. GTCs are crucial for the prevention and treatment of human breast cancer owing to their potential to target the molecular cues responsible for breast cancer.

One of the worst malignancies in the world is hepatocellular carcinoma (HCC), an invasive liver cancer [109]. Numerous scientific and clinical investigations have looked closely at the anti-hepatocellular carcinoma properties of green tea catechins. Now it is known that EGCG can suppress the development and spread of hepatocellular tumors by inducing apoptosis, controlling autophagy, and acting as an anti-angiogenic agent [110]. EGCG has been shown to inhibit the growth of human HCC cells in in vitro studies using hepatic cancer cell lines by blocking the phosphorylation of the tyrosine-kinase receptor insulin-like growth factor 1 receptor (IGF-1R), initiating cell death by activating Caspase-9 and -3, suppressing Bcl-2, cyclooxygenase-2 (COX-2) and lipogenic enzymes, regulating the levels of VEGF and its receptor (VEGFR-2), NF-κB and ERK1/2, actuate AMPK (adenosine monophosphate-activated protein) and reactive oxygen species (ROS)-mediated membrane permeability [111]. The beneficial effects of GTCs intake were tested in various animal models, and its ability to suppress HCC growth, proliferation, and trigger apoptosis has been confirmed. It was hypothesized that GTCs prohibit hepatocyte stem cell growth, excitation of adenosine monophosphate-activated protein kinase (AMPK) protein inside the liver, and regulation of epigenetic expression [110,112]. Additionally, EGCG reduced the activity of the VEGF/VEGFR or IGF/IGF1R signaling axis, which in turn reduced the development of HCC xenografts [113,114]. Therefore, it is thought that GTCs, particularly EGCG, may be useful in preventing hepatic cancer.

Prostate cancer is one of the most common diseases affecting men and is currently a serious public health issue [115]. It is observed that the regular intake of green tea catechins (for 1 year) not only reduces the serum IGF-1, VEGF, and the target antigen for the prostate but also prevents the advancement of mature intraepithelial neoplasia to prostate cancer [116,117]. Recently, green tea has been utilized as a radiosensitizer for the effective treatment of prostate cancer via radiotherapy [118]. Numerous studies have suggested that GTCs hindered prostate cancer progression at various stages of the disease by reducing the expression of the prostate-specific antigen (PSA), the transcriptional activity of the androgen receptor (AR), the activation of chromatin proteins, and DNA methylation of genes [119,120]. Additionally, Gupta et al. (2000) showed that treatment of prostate cancer with various concentrations of EGCG (20, 40, and 80 µM) could pronounce notable effects on G0/G1 phase arrest and cause cellular death by upregulation of p21 waf1 expression [121]. Furthermore, the same group identified the cdk inhibitor-cyclin-cdk machinery as the relevant bio-molecular pathway for EGCG-instigated cell cycle overactivity and apoptosis. EGCG (5 µM) has also been noted to constrain cell migration and incursion via impacting lipid rafts to disable stimulation of the tyrosine protein kinase Met receptor [122]. The involvement of upregulated p53, p21, Bcl-2-associated X protein(bax), and cysteine protease-9 (CASP-9a) expression and DNA-binding protein inhibitor (ID2) downregulation was other also linked to prostate cell death [123,124,125]. The therapeutic efficacy of EGCG against prostate tumorigenesis was reported in transgenic adenocarcinoma of mouse prostate (TRAMP). The study results on mice illustrated that a 0.06% uptake of EGCG in drinking water for 28 weeks reduced prostate cancer proliferation and induced their death by down-regulating molecular mediators that support cancerous phenotype [126,127]. Based on the aforementioned research, EGCG can be used as an attractive chemo-agent for prostate cancer prevention.

The prevalence of gastrointestinal (GI) cancers poses a grave threat due to their high risks and mortality in humans [128]. According to an epidemiological study, drinking green tea can lower the incidence of GI cancer, especially those pertaining to the esophagus and colorectum [129]. In fact, green tea’s role in the prevention of esophageal cancer was validated by a meta-analysis of clinical studies done in China [130]. Several works, both in vitro and in vivo, have demonstrated GTCs’ influence on gastric and colorectal cancer. Some of the effects on tumor vascularization, DNA methylation, reduced tumor cell growth and proliferation, and programmed cell death were all postulated. It is postulated that EGCG hinders cancerous cell formation and invasion by disrupting the molecular signals of cancer cells. Moreover, EGCG was proposed to prompt G1 cell phase arrest, Glycogen synthase kinase 3 (GSK-3) and protein phosphatase 2A(PP2A) phosphorylation and decrease epidermal growth factor receptor (EGFR), human epidermal growth factor receptor 2 (HER2) activation and wingless/Integrated (Wnt)/catenin signal pathways [131,132]. According to other studies [133,134]. EGCG can also end up arresting the cell cycle by impairing the transcription factor and signaling pathways such as (activator protein (AP-1), Akt, ERK1/2, p38, MAPK, NF-B and also cause a reduction in the level of essential anti-apoptotic proteins. Similarly, Umeda et al. [135] found that, at a physiological dose of 1 M EGCG, EGCG stimulated G2/M cell cycle arrest and activated laminin (receptor) mediated myosin phosphatase. Adachi and colleagues conducted investigations with the goal of treating diseases related to obesity. These studies revealed that EGCG reduced the development of colorectal cancer cells by decreasing the expression of a variety of Receptor Tyrosine Kinas (RTK)-mediated cell signaling pathways [136,137]. Activating Bax and Caspase-3, lowering the levels of proliferation protein cyclin D1 and regulator protein Bcl-2 in the NF-B channels, and inhibiting the G0-G1 phase have all been observed in esophageal cell lines [138]. The production of VEGF and VEGFA, as well as the activation of the VEGF/VEGFR axis, have also been reported to be inhibited by EGCG [139] in colorectal cancer cells. EGCG has been shown to modulate the epigenome to regulate the methylation degree of the transcriptional regulator Retinod X Receptor (RXR) promoter [140]. Animal, model-based studies validated the anticancer effects of EGCG on N-nitro methylbenzylamine (NMBA)-induced rat esophageal carcinogenesis via targeting cyclin D1 and affecting COX-2 expression [141]. Additionally, EGCG oral dosages of 5, 10, and 20 mg/kg led to a reduction in colorectal cancer cell volume. This resulted from EGCG’s antagonistic effect on the expression of Notch signaling. Additionally, researchers examined the effects of intraperitoneal injection of 1.5 mg/d of EGCG on angiogenesis and tumor formation in naked mice bearing stomach cancer xenografts. Possible causes included reduced VEGF and CD31 expression [139,140,142]. In a colon carcinogenesis model, EGCG suppressed the activation of the IGF/IGF1R axis and the VEGF/VEGFR axis, according to [113,143]. Overall, the studies mentioned above show that GTCs hold a prominent position in gastrointestinal tract carcinogenesis prevention and cure.

About 25% of cancer-related deaths are caused by lung cancer. Lung cancer claims more lives each year than colon, breast, and prostate cancers combined [144] (https://www.cancer.org/cancer/lung-cancer/about/key-statistics.html, accessed on 12 January 2023) Various results on the effect of GTCs consumption in the suppression of chemically induced lung tumors in transgenetic rodents are available. Similar to the aforementioned cancers, GTCs have been found to have both curative and preventive effects on lung cancer through obstruction of key protein kinases and regulation of molecular signals and gene expression of cyclin D1, Bcl-2, p21, p53, VEGF, Bax COX-2, Caspase-3, -7, and -9 [145,146,147]. Lu and colleagues [148] corroborated the hindering ability of GTCs to be more effective than caffeine against lung carcinogenesis in experiments with 4-(methylnitrosamino)-1-(3-Pyridyl)-1-Butanone (NNK)-induced A/J mice, administering 0.5% Polyphenon E (65% EGCG) or 0.044% caffeine as dietary fluid for 52 weeks. It was also claimed that EGCG could significantly inhibit phosphorylation [148]. Similarly, H1299 and Lu99, types of non-small cell lung cancer (NSCLC), when treated with 50 and 100 µM EGCG. were shown to suppress metastasis of lung carcinoma. For the probable mechanism of inhibition of epithelial-mesenchymal transition (EMT), phosphorylation of matrix metalloproteinases-9 (MMP-9) with tyrosine and focal Adhesion Kinase (FAK) activities were suggested [149,150,151]. The effect of EGCG (0–20 µM) on H1299 cells has shown the affinity of EGCG attachment with rat sarcoma (Ras)-GTPase activating protein src Homology-3 (SH3) domain-binding protein 1 (G3BP1), production of ROS and cancer cell apoptosis [152,153]. Treatments with 10–100 µM EGCG have shown remarkable effects on lung carcinogenesis and its spread by blocking multiple protein receptors, nicotine-driven migration, and differentiation of Non-small cell lung cancer (NSCLC) cells [154,155]. Likewise, treatment of NSCLC with EGCG (0–120 µM) has been shown to disrupt the enzyme expression of telomerase along with Caspase-3 and -9 activities [156]. As a result, catechins can function as targeted agents to stop or slow the development of lung tumors. As summarized in this section, GTCs’ anticancer effects are indeed diverse and well investigated, and authoritatively confirmed. Figure 3 summarizes the various signaling pathways and molecular targets of GTCs behind the anticancer activity of GTCs.

## 4. Anticancer Activity of Catechins Combined with Natural Compounds

With the proven anticarcinogenic activity of GTCs gaining eminence, various combinatorial approaches, where GTCs were combined with other anticancer drugs/natural compounds, have also been tested. Generally, natural compounds hold a clear advantage over drugs for reasons such as their natural availability, abundance, and bio-safety. The combined effect of tea catechins (EGCG) with other natural compounds have shown a complementary effect in exerting anticancerous property due to improved bioavailability of pro-apoptotic metabolites. The various mechanisms of catechins in enhancing anticancer effects when combined with other natural substances are compiled in Table 1. For example, the combination of curcumin and catechins has shown better anticancer effects on a wide range of cancer cells [157,158,159,160,161,162]. Other bioactive substances, such as 6-gingerol, pan-axadiol, sulforaphane, and pterostilbene, have been able to exhibit mutual interactions with GTCs like that of curcumin [163,164,165,166]. Natural components such as Piperine and genistein are others that have been observed to enhance EGCG bioavailability by hindering glucuronidation and MRP-mediated efflux, thereby controlling the mobility of EGCG within the gastrointestinal region [167,168]. Furthermore, it has been reported that combining GTCs with ascorbic acid, sucrose, quercetin, and carbohydrate has enhanced the bioaccessibility and intestinal absorption of catechins [169,170]. It has been demonstrated that luteolin, in combination with EGCG, has enhanced anti-lung cancer properties via the increased activation of cancer-suppressor p53 protein [171]. Similarly, Papi and colleagues concluded that the mixtures of vitexin-2-oxyloside (40 g/mL), raphasatin (5 g/mL), with EGCG (10 g/mL) can inhibit colon cancer cell proliferation and trigger its death via the mitochondrial-mediated pathway [172]. More recent work on integrating a silibinin mixture with EGCG has shown a more promising way to target cancer endothelial cells [173].

## 5. Anticarcinogenic Antioxidant/Prooxidant Effects of Catechins

The specific chemical structure of polyphenols found in green tea (the presence of a minimum of five hydroxyl groups) has a substantial impact on antioxidant capacity [183,184,185,186,187,188]. This is because green tea contains at least five hydroxyl groups. The di/tri-hydroxy structure of the B and D rings [185], as well as the meta-5,7-dihydroxy group at the A ring [189,190], help in the chelation of transition metal ions [183,184]. For synthesizing any novel anticancer molecule(s) based on the structure of catechins, (i) the molecule should be of epicatechin type; (ii) it should have as many galloyl moieties because this will increase the number of hydroxyls that can bind to DNA and Cu(II) and reducing it to Cu(I) [183]. Nevertheless, in certain cases, they might also exhibit prooxidative effects [183,184,191]. The actual anticancer mechanism is through the regulation of catechins within an intracellular pool of nitro-oxidative stress [192]. This is the reason why polyphenolic chemicals, especially catechins at high dosages, can also result in adverse effects [193] because catechins are a type of antioxidant. Hence, at high dosages, induction of prooxidative stress as well as oxygen damage to the constituents of the cell result. Close connectivity between inflammation and prooxidative action also exists [193]. Catechins have qualities that are both antioxidant and prooxidant, and both properties are dependent on the same variables (OH groups) inside the molecule [183]. Oxidation of polyphenols results in reactive oxygen species and electrophilic quinones that can damage cells [194,195]. Thus, this prooxidative nature of catechins should also be seriously taken into account since it plays a vital role in the etiopathogenesis of degenerative disorders leading to cancer [193]. Figure 4 summarizes the anticancer activity of catechin exercised via its antioxidant activity.

## 6. Challenges, Future Recommendations, and Conclusions

One of the major challenges facing the biomedical application of GTCs is their low bioavailability. Certain progress has been made in this related aspect; however, clear resolutions are far from being achieved. One of the key strategies towards enhancing the antitumor potential of GTCs for improving their bioavailability was through combining catechins with other phytochemicals and anticancer compounds through nanostructure-based delivery systems and molecular modifications. Standalone GTCs, in spite of their high reputation, have their own limitations, this is where combinatorial approaches take lead positions. Combination therapies with catechins can exert synergistic effects through interaction of catechins with bioactives or drugs, leading to reduction of the side effect of conventional chemotherapy and enhanced anticancer potential. Chemically modifying EGCG or other catechins, using nanotechnological as well as drug delivery approaches has a wider perspective in improving its anticancer effect [40]. There is a lot to expect from GTCs in the field of anticancer research. This review prompts more research attention in these areas. There are abundant in vitro investigations; more in vivo investigations are required to understand and improvise the available resources from GTCs for human benefit.

Thus far, binary combinations have been worked out with GTCs, trinary or ternary composites, combining two or three more versatile and dynamic properties of natural compounds (for example, EGCG/cur/lovastatin), need be attempted since the success rates in overcoming the limitations of catechins though such multiple combinations can be relatively higher. Moreover, through the course of the review, we could find only a handful of research articles that reported GTCs/natural compound combinations, with most studies localized around the use of curcumin. Nature has more to offer than just curcumin, and there are diverse natural antioxidant compounds reputed for their strong anticancer activity (for example, resveratrol, indole-3-carbinol, vitamin D, chrysin, celastrol, and betulinic acid); this review urges more expansion through attempting many other available natural options. Additionally, EGCG is, for the most part, worked on GTC; the other green tea catechins also possess unique properties, and combinations of two or more catechins with natural compounds could exhibit more versatile anticarcinogenic properties.

GTCs’ systemic action mechanism in the human body is far from understood; it is necessary that we obtain a more comprehensive understanding of EGCG-mediated anticancer mechanisms by exploring whether the relative molecules are direct targets for EGCG and then substantiate the in vivo relevance of the reported mechanisms. Targeted delivery by attaching specific ligands to nanoparticles is another area that has not been looked into in detail. The clinical translation of GTCs, their pharmacokinetics, and drug delivery aspects need more investigation [40]. The results from clinical trials with EGCG are not all positive. A study performed on 49 patients with various tumors reported no major antitumor responses when using GTE. The catechin concentrations and administration methods, as well as other parameters, need to be optimized to obtain a clear picture. More importantly, the safety aspects and the side effects of GTCs should be investigated in clinical trials, and the related mechanisms should be theoretically/practically demonstrated.

The various anticarcinogenic achievements of GTCs were reviewed, and the progress made in anticancer therapies was summarized. The combinatorial enhancement of the ant-cancer attributes of GTCs, through integration with renowned natural compounds invested with anticancer properties, has been specifically highlighted, and the future directions proposed.

## Figures and Tables

**Figure 1 molecules-28-02151-f001:**
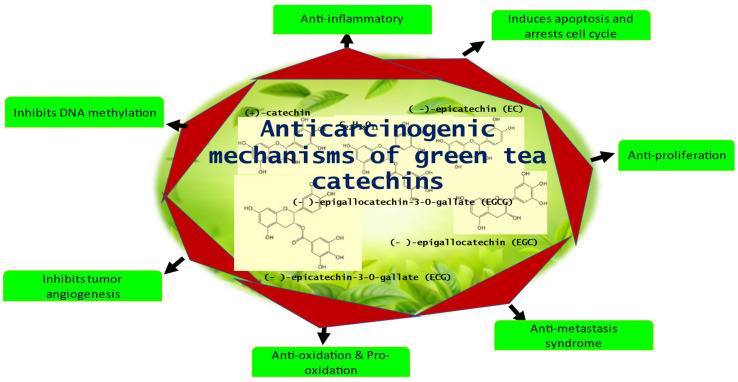
The Anticarcinogenic mechanisms of GTCs.

**Figure 2 molecules-28-02151-f002:**
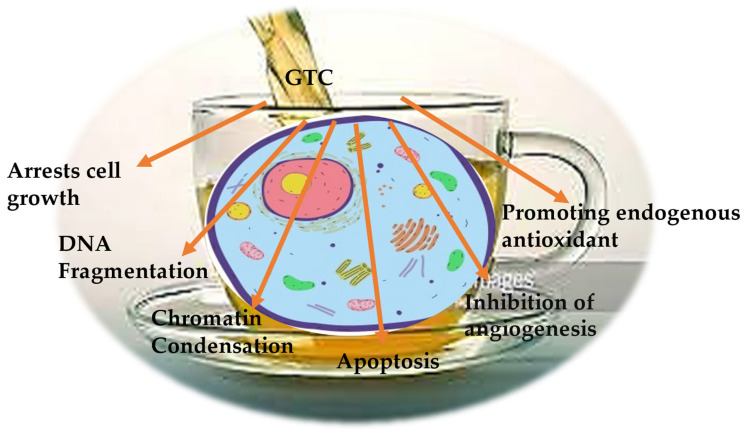
The anticancer mechanisms of GTCs.

**Figure 3 molecules-28-02151-f003:**
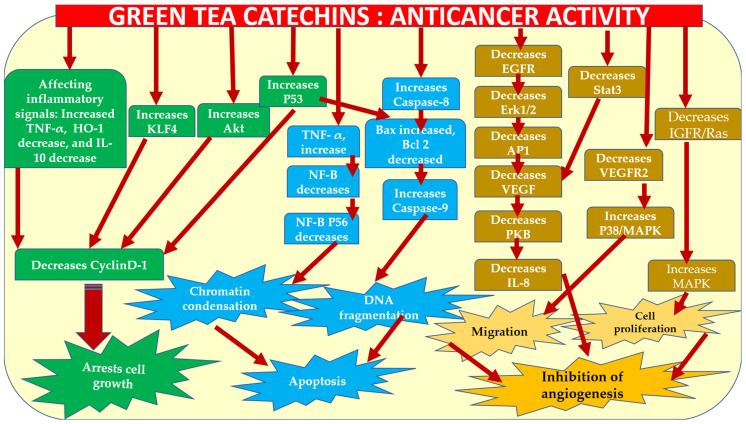
The molecular targets and signaling pathways behind the anticancer activity of GTCs. TNF—tumor necrosis factor; HO-1—Heme oxygenase-1; IL—interleukin; KLF4—Kruppel-like factor 4.

**Figure 4 molecules-28-02151-f004:**
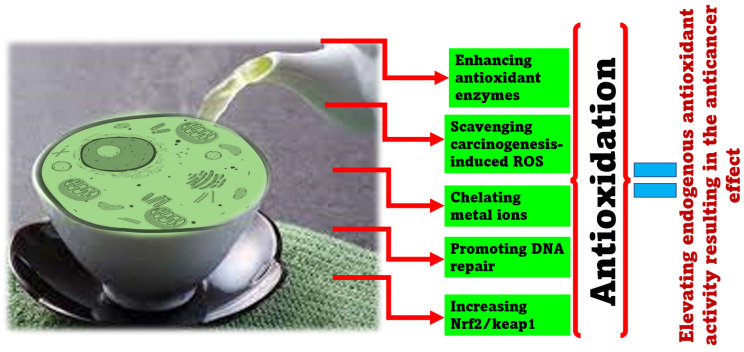
The molecular targets and signaling pathways are affected via the antioxidant activity of GTCs resulting in the anticancer effect.

**Table 1 molecules-28-02151-t001:** Anticarcinogenic effects of GTCs when combined with antioxidant-rich anticancerous natural compounds.

GTC Component	Natural Compounds	Cancer Type	Anticarcinogenic Activity	Molecular Mechanism	References
EGCG	Curcumin	MCF-7 breast cancer	Inhibition of cancer cell growth and induction of apoptosis	Activation of caspase-dependent apoptosis, inhibition of P-gP pump function	[160]
EGCG	Curcumin	MDA-MB-231 breast cancer	Inhibition of cancer cell growth	G(2)/M-phase arrest. Decreased VEGFR-1 protein expression in tumours	[159]
Epicatechin	Curcumin	HL-60 myeloid leukaemia	Inhibition of cancer cell growth	Cell cycle arrest at S phase	[174]
EGCG	Curcumin	Normal, premalignant and malignant oral cells	Inhibition of cancer cell growth	Cell cycle arrest at G1 phase and S/G2M	[158]
EGCG	Curcumin	A549 and NCI-H460 lung cancer	Inhibition of cancer cell and tumour growth	Cell cycle arrest at G1 and S/G2 phases via cyclin D1 and cyclin B1 inhibition	[162]
EGCG	Curcumin	PC3—human prostate cancer	Inhibition of cancer cell and tumour growth	Arrests S and G2/M phases by upregulated expression of p-21	[157]
EGCG	Curcumin and lovastatin	SKGT-4 and TE-8 esophageal cancer	Inhibition of cancer cell and tumour growth	Suppression of mitotic signal transduction pathway through Phosphorylation/dephosporylation of /Erk1/2, c-Jun and COX-2	[161]
EGCG	Silibinin	Non-small-cell lung cancer cells	Inhibition of angiogenesis and cell migration of endothelial and lung tumour cells	Antiangiogenic activity via VEGF, VEGFR2, and miR-17–92 cluster and miR-19b cluster	[173]
EGCG	Luteolin	esophageal cancer cell lines—TE-8 and SKGT-4	Inhibition of growth and induction of apoptosis of cancer cells and tumour	induction of p53-dependent apoptotic pathways	[171]
EGCG	6-Gingerol	1321 N1 and LN18 glioma cells	Inhibition of cancer cell growth and induce apoptosis	Apoptosis induction by activated Caspase-3 and Annexin-V FITC/PI	[166]
EGCG	Panaxadiol (PD)	HCT-116 and SW-480 human colorectal cancers	Inhibits cancer cell growth and induces apoptosis	Cell cycle arrest at G1 and G2/M	[163]
EGCG	Sulforaphane (SFN)	PC3 prostate cancer	Arrests cancer cells	Inhibition of genes in AP-1 pathway inhibits cell proliferation, differentiation, apoptosis, angiogenesis and tumour invasion	[165]
EGCG	Pterostilbene	MIA PaCa-2 and PANC-1 Pancreatic cancer cells	Inhibits growth of cancer cells and induces apoptosis	Induction of cell cycle arrest and cell apoptosis	[164]
EGCG	Paclitaxel (Taxol)	Breast cancer cells (4T1, MCF-7, and MDA-MB231)	Inhibits growth of cancer cells and induces apoptosis	JNK phosphorylation and cell death	[175]
EGCG	taxane (i.e., paclitaxel and docetaxel)	Human Prostrate cancer cells PC-3ML cells	Inhibition of growth and induction of apoptosis	Induction of cell cycle arrest and cell apoptosis	[176]
EC epicatechin	Panaxadio	HCT-116 human colorectal cancer cells	Inhibition of cancer cell growth	Mechanism not reported	[177]
EGCG	Cisplatin/ tamoxifen	1321N1 cells and U87-MG cells/ biliary tract cancer cells	Cytotoxicity on Cancer cells	Inhibition of telomerase and induction of cell cycle arrest	[178,179]
EGCG	doxorubicin	Drug-resistant KB-A1 cells	Cytotoxicity on Cancer cells	Modulating P-glycoprotein efflux pump	[180]
EGCG	doxorubicin (DOX)	human oral epidermoid carcinoma (KB-A-1)	In vivo reversal of doxorubicin resistanceby (-)-epigallocatechin gallate in a solid human carcinoma	Induction of cell apoptosis	[181]
EGCG	Vitexin-2-o- xyloside, Raphasatin	Colon cancer cells—LoVo and CaCo-2; Breast cancer cells—MDA-MB-231 and MCF-7	Inhibition of cancer cell growth and induction of apoptosis	Arrests Cell cycle at G0/G1 phases and activating ROS- mediated mitochondrial apoptotic pathways	[172]
EGCG and ECG	DOX	Chemoresistant hepatocellular carcinoma (HCC) cell line BEL-7404	Inhibition of cancer cell and tumour growth.	Inhibition of P-glycoprotein efflux pump	[182]

Abbreviations: EC—(−)-epicatechin; EGC—(−)-epigallocatechin; ECG—(−)-epicatechin-3-gallate; EGCG—(−)-epigallocatechin-3-gallate; ROS—reactive oxygen species; JNK—Jun N-terminal kinase; AP1—activating protein; VEGF—vascular endothelial growth factor; COX—Cyclooxygenase; Erk—Extracellular signal-regulated kinase; FITC/PI—fluorescein isothiocyanate; PI, propidium iodide; miR—microRNAs.

## Data Availability

Not applicable.

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
