# Peer review of "Anticancer Therapeutic Effects of Green Tea Catechins (GTCs) When Integrated with Antioxidant Natural Components"

_molecules, 2023, doi:10.3390/molecules28052151_

Round 1
Reviewer 1 Report
The review aimed to report the biological benefits of catechins in green tea and highlighted on its anticancer effects when it is in combination with various chemical compounds from natural sources.
- The aim is good and sound.
- Title: I prefer to be more general as this title pointed more on the combinatorial anticancer effect.
- A list of abbreviations is recommended to be included in the review.
- Some words need correction: line 73 (3- oils); line 348 (ncreased).
- Some words should be written italic as "via" (lines: 81, 171, 347, 350, 379); "in vivo" & "in vitro" (lines:169, 413).
- Definition of abbreviations as footnotes under the table and figures is recommended.
- One reference is missed (line 140).
- Table 1: is Lovastatin from natural source?; in the last row "Chemoresistant hepatocellular carcinoma.." needs more clarification as it is in the column of natural compounds; addition of concentration or doses of the natural compounds correlated to the anticancer activity is recommended; also addition of the class of compounds.
Author Response
The review aimed to report the biological benefits of catechins in green tea and highlighted on its anticancer effects when it is in combination with various chemical compounds from natural sources.
Ans. We would like to thank the reviewer for the encouraging comments and for the valuable suggestions, we have revised the manuscript according to your valuable comments. We provide a point by point response to the queries below.
- The aim is good and sound.
- Ans. Thank you.
- Title: I prefer to be more general as this title pointed more on the combinatorial anticancer effect.
- Ans. We have modified the title.
- A list of abbreviations is recommended to be included in the review.
- Ans. in MDPI style there is no provision for a separate list, they insist we mention the abbreviation in text at the site of first mention. We have now revised making sure we have done this. Thank you.
- Some words need correction: line 73 (3- oils); line 348 (ncreased).
- Ans. Sorry about that, corrected now.
- Some words should be written italic as "via" (lines: 81, 171, 347, 350, 379); "in vivo" & "in vitro" (lines:169, 413).
- Ans. Yes you are right, we have corrected these.
- Definition of abbreviations as footnotes under the table and figures is recommended.
- Ans. Added in revised version under tables and figures. Those not explained in text have been added. thank you
- One reference is missed (line 140).
- Ans. Taken care of .
- Table 1: is Lovastatin from natural source?; in the last row "Chemoresistant hepatocellular carcinoma.." needs more clarification as it is in the column of natural compounds; addition of concentration or doses of the natural compounds correlated to the anticancer activity is recommended; also addition of the class of compounds.
- Ans. Lovastatin is a naturally occurring compound found in low concentrations in food such as oyster mushrooms, red yeast rice, and Pu-erh. Sorry about the last row, modified it. The table is already an exhaustive one focusing on specific aspects we wish to highlight, so we stick to the current flow. Thank you for your understanding.
Reviewer 2 Report
Synergistic anticarcinogenic effects when green tea catechins (GTCs) are combined with other antioxidant rich natural compounds. Living in an age of inadequacies, combinatorial approaches are gaining momentum, GTCs have progressed much, yet there are insufficiencies which can be improvised when combined with natural antioxidant compounds.
The title and content of the article represent a topic of real interest worldwide. Most of green tea's anti-tumor properties are catechin-regulated, with EGCG having strongest impact, EGCG has the highest inhibitory activity, followed by ECG. Moreover, combinations of catechins have shown enhanced anti-tumor activity than isolated EGCG due to their combined effect. The subject of the study is topical with real interest for the future.
The introduction of the article presents originality by proposing a topic with a huge academic potential.
The bibliographic data inserted along the article presents a qualitative chronology. The subject of the article represents a true scientific revolution in its field.
The material and methods section of the article presents a quantitative and qualitative exposition of the research plan, respectively a good reproducibility in order to develop other studies with this theme. I consider it necessary to develop new studies on this subject and implement them on a population scale.
The results of the article present a logical and chronological exposition outlining qualitative aspects of the benefitanticarcinogenic effects when green tea catechins (GTCs) are combined with other antioxidant rich natural compounds.Catechins have qualities that are both antioxidant and pro-oxidant, and both properties are dependent on the same variables (OH groups) inside the molecule. The figures and tables keep a specific chronology throughout their exposition, presenting qualitative aspects related to the subject of the article.
The topic of the article is a real interest for the future with major importance in this field. I consider it necessary to develop new studies on this subject and implement them on a population scale. The article presents an important research point with an optimal linguistic exposition, having an exponential potential for the future. This present article is written in a clear and concise manner.
The article presents originality, with an optimal literary exposition, representing a topic of real interest for the future with objective results at the research level. The article represents a launching platform in its field and from the point of view of the characteristics it is included for publication.
Author Response
Synergistic anticarcinogenic effects when green tea catechins (GTCs) are combined with other antioxidant rich natural compounds. Living in an age of inadequacies, combinatorial approaches are gaining momentum, GTCs have progressed much, yet there are insufficiencies which can be improvised when combined with natural antioxidant compounds.
Ans. We would like to convey our heartfelt thanks to your reviewer for the appreciation and motivating comments given, we are very much encouraged.
The title and content of the article represent a topic of real interest worldwide. Most of green tea's anti-tumor properties are catechin-regulated, with EGCG having strongest impact, EGCG has the highest inhibitory activity, followed by ECG. Moreover, combinations of catechins have shown enhanced anti-tumor activity than isolated EGCG due to their combined effect. The subject of the study is topical with real interest for the future.
Ans. Thank you
The introduction of the article presents originality by proposing a topic with a huge academic potential.
Ans. Thank you very much
The bibliographic data inserted along the article presents a qualitative chronology. The subject of the article represents a true scientific revolution in its field.
Ans . thanks a lot
The material and methods section of the article presents a quantitative and qualitative exposition of the research plan, respectively a good reproducibility in order to develop other studies with this theme. I consider it necessary to develop new studies on this subject and implement them on a population scale.
Ans. Thank you
The results of the article present a logical and chronological exposition outlining qualitative aspects of the benefitanticarcinogenic effects when green tea catechins (GTCs) are combined with other antioxidant rich natural compounds.Catechins have qualities that are both antioxidant and pro-oxidant, and both properties are dependent on the same variables (OH groups) inside the molecule. The figures and tables keep a specific chronology throughout their exposition, presenting qualitative aspects related to the subject of the article.
Ans. thank you.
The topic of the article is a real interest for the future with major importance in this field. I consider it necessary to develop new studies on this subject and implement them on a population scale. The article presents an important research point with an optimal linguistic exposition, having an exponential potential for the future. This present article is written in a clear and concise manner.
Ans. thanks a lot for taking time to pen down every aspect of the manuscript in recognition.
The article presents originality, with an optimal literary exposition, representing a topic of real interest for the future with objective results at the research level. The article represents a launching platform in its field and from the point of view of the characteristics it is included for publication.
Ans. We are so encouraged, thank you for your timely and prompt review. thanks again.